# Immersive Haptic Authoring: Creating Haptic Experiences On-Demand within VR through Direct Manipulation

Institution

Anonymous for Submission†
Institution

Anonymous for Submission‡
Institution

Anonymous for Submission§
Institution

## ABSTRACT

This paper presents *immersive haptic authoring*, a novel way to author haptic experiences within VR. Designing haptics to create rich VR experiences is becoming increasingly common, yet designers lack the tools to iteratively design, experience, and test such experiences easily. Designing haptics using desktop tools is a slow, distracting and unintuitive process. To address this, we introduce *immersive haptic authoring*, a new approach that allows the haptic designer to map, program, and modify haptic experiences *within VR* through direct and spatial manipulation, which can support rapid iteration and exploration of new haptic experiences. In this paper, we develop a system to demonstrate the concept and present insights from a qualitative evaluation session with experts. The result suggest that our approach has many benefits including better design exploration, time efficiency, and provides immersive design interactions, which together provide us insights for the future of VR haptics authoring tools.

**Index Terms:** Human-centered computing—Human computer interaction (HCI)——

## 1 INTRODUCTION

Recent advances in consumer virtual reality (VR) devices have shown great promise in creating visually immersive environments. Virtual reality is currently a largely visual and auditory experience that allows rich and immersive scenarios to be created. However, this immersion is broken when interacting with objects without the *sensation of touch*. To address this, many researchers have explored haptics interactions to enrich VR experiences. Haptic interactions enabled by vibration [2, 22], force [4], temperature [22], and airflow [16, 17] can greatly enhance the immersion of VR experiences, and promises many application domains including entertainment [1], education [35], and training [13].

However, the existing research mainly focuses on *devices*, with less focuses on the *authoring process* — i.e., how we can support haptic designers to develop such haptic experiences. For example, the common practice to design such haptic interactions still largely rely on traditional game programming environment (e.g., Unity or Unreal) or desktop GUI systems (e.g., BHaptics editor). These practices produce three key limitations:

1) **Context-switch between Editing and Experiencing**: The current practice has a separation between editing in the real world but experiencing in the VR world which creates an undesirable division in the implementation and evaluation process.

---
*e-mail: email@site.org
†e-mail: email@site.org
‡e-mail: email@site.org
§e-mail: email@site.org

2) **Unintuitive authoring systems**: These workflows often require either directly coding an affect, or the use of 2D GUI panels, which produces a steep learning curve that hinders novice users and produces unintuitive authoring relationships such as changing a direction of force by needing to define a mathematical concept such as a quaternion rotation equation.

3) **Large time overhead**: Every change to the haptics, even very minor adjustments, requires the time to create the edits, compile and load the software, wear any necessary hardware, and go to the VR location to experience the result which produces a practical limitation on the number of prototype iterations that can be done and incentivises only high value changes.

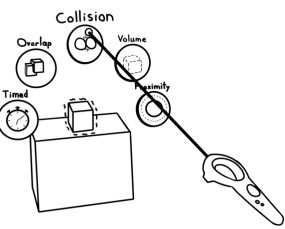

Figure 1: Example VR authoring interaction

To address these problems, this paper introduces *immersive haptic authoring*, a novel way of authoring the haptic experiences within VR. Immersive haptic authoring allows the user to define, modify, and test a haptic experience within the VR environment, which has the following three key features:

1) **Context-based designing:** All of the interactions and authoring can be done within the VR enviornment being designed, in-turn helping the designer constantly reference the context being considered. For example, it could be easier to brainstrom haptics for a medical application while being immersed in a surgery simulation compared to imaging it on paper or otewise.

2) **Direct and spatial manipulation:** Being immersed in the VR enviornment and with in-situ access to the hardware, the designer can directly and spatially manipulate behaviours and controls, offering more avenues for exploration. For example, a designer can test many variations of haptic experiences tied to body movement while being immersed in the simulation.

3) **Fast iteration through author-experience integration:** The immersive nature of such an authoring process would enable designers to create and edit haptics without needing to context switch between software programmin and testing using hardware.

To demonstrate our ideas, we designed and implemented a prototype system which supports the haptic experience design of the controllers vibrating, a vibration vest that covers the entire front and back torso, and a pair of vibrating bracers.

To investigate the strengths and limitations of immersive haptic authoring, we conducted a preliminary qualitative study with six expert designers who provided insight through interview driven feed-

back after experiencing our prototype system. Our results highlight the impact immersive haptic authoring can have on designer motivation, authoring conceptualization, and presence in the VR design, the usefulness of physical manipulation as a design interaction, and the expectations designers have for the authoring tool implementation. Informed by our results we discuss the role we believe immersive haptic authoring should play in designing a VR haptic experience, the effect and expectations of this approach at both a theoretical and practical level, and what the next step in immersive haptic authoring research will involve.

In summary, we make the following four contributions:

1. We introduce a novel concept, immersive haptic authoring, informed by past research in VR-based haptics.

2. A prototype system that illustrates immersive haptic authoring

3. Insights from a user study that reflects on the potential benefits and limitations for in-situ haptic editors.

4. A set of immersive haptic authoring themes that future research should explore further.

## 2 RELATED WORK

Our work draws on literature related to two main topics: haptics in VR and VR-related authoring tools. Our work is informed by these topics and discusses how an integration of these two can offer designers a new opportunity to engage in the design of VR-based haptic experiences.

### 2.1 VR Haptic Devices

Effective haptic feedback promises to enrich VR experiences. In the literature of HCI, researchers have explored many different approaches to generating haptic sensations including force feedback [16, 18, 19, 38], temperature adjustment [6, 15], chemical application [5, 29], environment manipulation [20, 33, 36], and electro-muscular stimulation [27, 28].

In terms of the body parts, most commonly haptics are placed in handheld devices [16, 32, 36], but haptics on other areas of the body are also widely researched. Research has looked at placements on the head [17, 26], arms [19], legs [39], and torso [10, 23, 24], and consumer electronics are also now seeing more expansive haptic areas including full hand [4, 31] and torso [2, 3] feedback. The expansion of areas to experience haptics provides unexplored opportunities for richer and more immersive haptics, so our approach considered haptics that could be desired anywhere on the body.

As we can see, there are a number of different haptic approaches, the most commonly used haptic sensation is *vibro-tactile feedback* [11], especially in commercial devices. Due to its flexibility and inexpensive actuation, vibro-tactile feedback is used to indicate contact [25], texture [34], stiffness [30], movement [9], and more. The goal of this paper is not to invent a new haptic sensation, but to explore the authoring experience for existing haptic devices, thus we specifically choose vibro-tactile feedback (for both hand-held controller and body-based wearable jackets) as our main haptic medium.

For vibro-tactile sensation, designing haptic experiences means that changing the parameter of vibration such as how to vibrate, when to vibrate, and where to vibrate to simulate many different virtual objects (e.g., fire, rain drops, electric shock, object shaking, surfaces, etc). However, even such a simple authoring, the existing haptic editing process is still largely limited with the very minimal authoring support, as we discuss next.

### 2.2 Traditional Haptic Authoring

Many modern haptic hardware devices come with supporting software to program functionality and patterns for use. Haptic jackets are of particular note as it involves addressing a large number of haptic elements together to form a cohesive haptic experience [2, 10].

HFXStudio [7] looked at how to allow authoring of haptic experiences beyond replicating physics driven modeling and allow custom interactions to take place. This approach used a level abstraction where the tool allowed the authoring of a desired result which was then fulfilled as best as possible by the available hardware present. The pilot study of this system found that both the egocentric and allocentric effects authored were considered intuitive, the interface control had two large issues. Firstly, differences in navigation between difference systems caused confusion in users, and secondly, the body selection tool only allowed selecting vertex-by-vertex making it cumbersome.

These type of interface issues can be handled by our system since navigation within the scene is done intuitively by physically moving one's body, and more complex interaction modalities become available from the natural 3D representation of object.

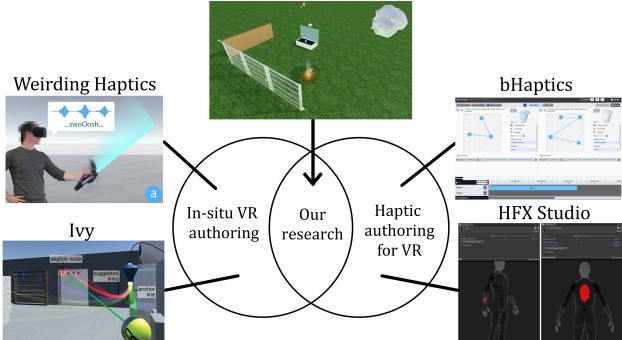

Figure 2: Our focus in relation to related work [2, 7, 9, 12]

### 2.3 In-Situ Authoring Tools for VR

The approach of using real-time in-situ authoring in VR has been found beneficial in a verity of contexts including programming [12], scene editing [37], and haptics [9]. These works explore how designers can use VR to interact with objects and produce good authoring experiences from a verity of contexts.

Ivy [12] looked at how in-situ authoring tools could be applied to programming functionality for smart objects from a VR environment. Using a set of nodes such as triggers, filters, and converters, with links between them, a user is able to author programming scenarios such as activating a museum exhibit in a room when enough people stood on a pressure pad. The researchers identified four guidelines when creating spatially situated visual programming environments which we have re-imagined for application with haptic authoring. A quick overview of the four guidelines are 1) Maintain spatial relationships: Have objects and abstractions such as logic coupled using spacial association to help users associate related logic. 2) Facilitate spatial interaction: Use actions such as head or body motions and familiar physical manipulations. 3) Embrace physical properties: Use symbolic representations to give physical form to logical constructs. 4) Expose minimally sufficient information: Avoid visual clutter and only include essential details as the being in VR may place a higher strain on a designer's attention. In the concept section of this paper, we describe a proposed set of guidelines with these as inspiration but that focus on the unique challenges and opportunities of a haptic authoring system instead of a logical programming system.

Most closely related to our work, Wireding Haptics [9] explores an on-demand haptics by using the user's humming sound. With that, the user can quickly and easily explore different haptic sensation for the vibration (e.g., sword swinging). However, the Wireding Haptics only focuses on the sound-based haptics, which limits the range of haptic experiences. In contrast, we propose the a general-purpose immersive haptic authoring process, which can support a variety of sensation for different purposes.

## 3 IMMERSIVE HAPTIC AUTHORING SYSTEM

An immersive haptic authoring system is one that allows a designer to create and iterate a haptic experience while remaining with the environment the user will ultimately experience it in.

### 3.1 Motivation and Three Key Features

As virtual reality haptic interactions become more common and the desire for haptics in VR experiences increases, tools that support the unique challenges of authoring such interactions are needed. Our concept of an immersive haptic editor makes three main design considerations:

**1) Context-switch between Editing and Experiencing:** Programming the interactions for haptic devices requires editing parameter from a monitor based GUI. This can either be in the editor used to control the VR environment or a support device such as an Arduino which is often present to directly control the haptic hardware. This means much of the designing and editing is done outside of the VR scene it is supposed to be experienced in. While it is certainly not impossible to imagine or remember the VR scene the haptics are being placed in context with, it does require an additional load on the designer to continually think of the various aspects that will work in combination with the haptic sensation.

**2) Unintuitive authoring systems:** Designing haptics for virtual reality means the final interaction will occur in 3D space. While desktop based tools have become proficient at allowing navigation and editing in 3D work spaces, VR allows unique benefits to editor interaction techniques. These include things such as using depth in workspace use or selecting and manipulating objects kinetically.

**3) Large time overhead:** When designing a haptic experience for VR in a traditional desktop approach, it often requires a work flow similar to designing a prototype of the desired sensation, integrating it into the VR scene, putting on the necessary haptic hardware and VR headset, navigating to a location the haptic affect will be triggered from, testing the sensation, and finally removing the hardware and headset to make a new edit. The large amount of time a single iteration takes can have three effects. 1) A lack of refinement in the experience if there is insufficient time to iterate. 2) Reluctance to experiment in new ideas if there is insufficient time to iterate. 3) The delay between having the idea to implement and experiencing the result breaks up flows of thought. Through an immersive haptic authoring system, no overhead is needed for haptic prototyping as the design iteration process can immediately transition into testing and then directly back to iteration.

### 3.2 Hardware and Software

To demonstrate the concept of an immersive haptic authoring systems, we created an example editor that allows designers to create a small set of vibro-tactile haptic interactions from within VR. Our system was created in Unreal Engine 4.23 and was designed and tested with the HTC Vive and Oculus Quest headsets. The haptic hardware used included the built in rumble devices in the controllers, a bhaptic Tactsuit, and a pair of bhaptic Tactosy devices to provide the haptic feedback.

As an early prototype, our system is not exhaustive in its use of a variety of possible haptic hardware devices. In our current prototype system we focus on demonstrating examples that leverage vibro-tactile feedback as it is a flexible haptic sensation that can be used for a variety of haptics including indicating contact, representing texture, stiffness, movement, and more.

Objects and models in the scene will have one or more "haptic spaces" attached to them. Each haptic space tracks all relevant data needed to express a haptic interaction such as what intensity and duration to produce. The user has a set of tracked points on their model's hands and body. When one or more of the tracked points overlaps with a haptic space, the haptic space notifies a centralized governing monitor with the contents of it's haptic interaction. The monitor then determines if this is a valid state to produce haptics, translates the overlapping points from the character model to the indexes of the motors on the physical hardware device, and sends the required values the hardware needs to activate which in this case is the intensity and duration.

and what that interaction then feels like.

### 3.3 Process walk-through

In this section, we illustrate one concrete example of a designer authoring the haptic experience of an electric fence.

#### 3.3.1 Step 1: Ideation

The designer starts by getting an initial idea of the haptic experience they wish to author. This initial concept will be based on elements such as 1) the scenario they are creating haptics for, 2) previous experiences they have with the scenario in real life, 3) the haptic hardware they expect will be used, and 4) a concept of how the haptic sensations they can produce will correlate to the haptic experience a user will have. With elements like these in mind, the designer should have an initial concept goal they will be designing towards as they engage with this authoring approach. In our system the initial concepts of the scenarios have been pre-determined for participants to maintain consistency and so they can focus on concept considerations such as 1-4 above.

#### 3.3.2 Step 2: Creating the non-haptic components

Although not part of the immersive haptic authoring process, it is a pre-requisite that the designer has created and integrated the 3D visual assets and audio components related to the scenario they will be authoring haptics for into the VR world. This can be done through a desktop editor or immersive haptic authoring system for world building [8]. In our system the world already includes 3D assets for each scenario being designed for which have been placed in close proximity for ease of access. (see figure 3 (A))

#### 3.3.3 Step 3: Designer state

The designer now enters the VR space and begins authoring the haptic experience. If any of the authoring system's actions overlap with natural actions that will be take by an end user or there is extra information that should only be displayed during the authoring process, the system will need a way for the designer to indicate if it is being interacted with as a designer or an end user (for testing). An example of the overlapping of actions would be that users will touch objects to interact with them, but touching objects can also be a selection method for the author. Our system uses an "editor mode" button on the controller that toggles functionality between a designer state and an end user state. When in the editor mode, all haptic spaces that can be authored are made visible as a set of translucent coloured areas (see figure 3 (B)).

#### 3.3.4 Step 4: Selecting a trigger

With the system in an editing state, the designer will start by specifying what conditions the haptic experience should be experienced under. These conditions can be simple, such as to trigger when any part of the avatar overlaps with a space as we implemented in our system, or can be complex such as triggering based on proximity and only if some other variable or state is true, such as feeling rain

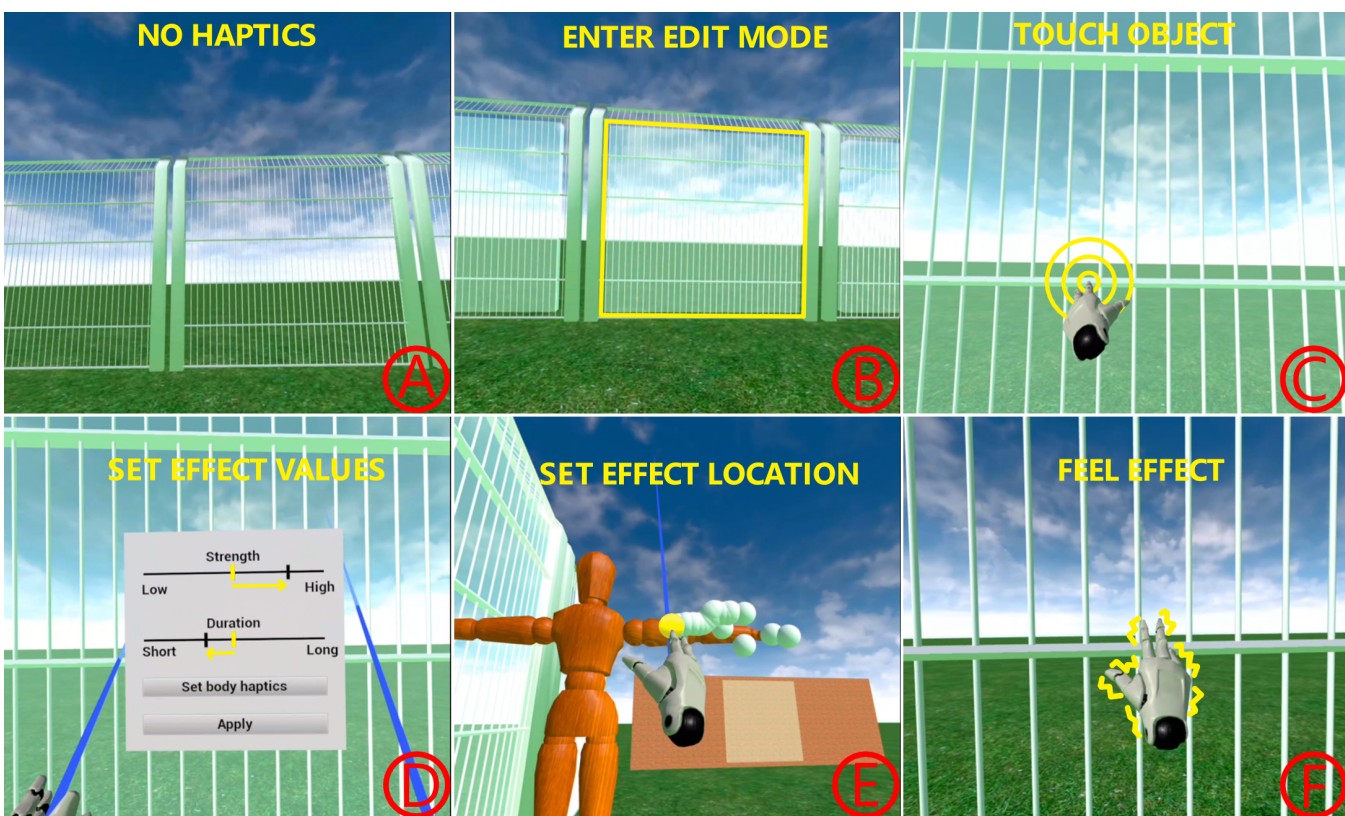

Figure 3: The process for creating haptics in our immersive haptic authoring system.

based on proximity to a tree but only if under a rain cloud. If the designer reaches out and touches one of the spaces with their hand, they begin editing the haptic interaction of that space through a menu that appears in front of them (see figure 3 (C)).

### 3.3.5 Step 5: Effects

With the trigger for the haptics selected, the designer must specify what haptic sensation should be felt to best evoke the haptic experience they are designing. There are two levels which this can be done at.

One option is the designer will edit the effect directly through the variables and values of a given haptic device to define a specific sensation. This can be seen in our system with the "strength" slider which sets the vibration intensity from %0 on the left to %100, and the "duration" slider which sets how long the sensation continues between 0 seconds on the left and 2 seconds on the right (see figure 3 (D)). This requires that the designer know what haptic hardware they are designing for and they will need to mentally convert their desired effect into a specific set of values that define a sensation.

The second option is the designer will edit the effect through a level of abstraction where they define a goal for the experience rather than the details to produce it. In our system, this can be seen when the designer uses the "select body" button and is able to indicate where on the body they would like the sensation to occur by drawing onto a mannequin with their index finger (see figure 3 (E)). In this case the designer is not specifying which motors to turn on, but instead what area of the body they wish to affect, and this can be interpreted regardless of if the haptic hardware is able to support producing a sensation in the given area.

### 3.3.6 Step 6: Results

Once the effect has been defined, the designer will save the interaction and move themselves from an editing state to the end user state which in our system is done by pressing the "editor mode" button again. When in the end user state, all additional elements unique to the design process are hidden such as in ours the translucent haptic space indicators to allow the designer to view and experience the haptic experience as a user would (see figure 3 (F)). This then allows the designer to interact with the scenario they are authoring and experience the results of their choices.

### 3.3.7 Step 7: Iteration

Finally, the designer reflects on the sensation they have produced and how successfully it invokes the experience they were trying to author. If after experimenting the resulting experience matches what they were trying to create they will stop. If the experience does match what they were trying to create though, it requires a reflection on what elements of the experience are wrong and what changes would be needed to produce a new iteration. The designer then return to step 3 to implement the new iteration.

## 4 USER STUDY

To better understand the concept and potential of immersive haptic authoring systems, we conducted a qualitative interview study with expert users. This section describes our study methodology.

### 4.1 Participants

We identified 10 individuals within the city with backgrounds in haptics or virtual reality, demonstrated either through their research work or multiple years of practical experience. Out of the 10 individuals contacted, 4 did not reply, 5 agreed to participate, and 1

forwarded the request to their Master's student who had studied VR under them and so was included in the study. This resulted in 6 participants to take part in the interview study.

While 6 participants is less that desired and is largely a practical limitation, it is still expected to produced meaningful results through open coding analysis [14]. We restricted recruitment to individuals within the city as participants needed to be at the research lab to have access to the required hardware, use the demonstration system in a stable environment, and have their interaction properly recorded. The background in design requirement was included so participants would have an understanding of the iterative process necessary in prototyping which is a core concept of this proposed approach to authoring. A background in VR or hapics was required because it is necessary to understanding how haptic authoring and VR environments could interact and how workflow may change. It is also expected that familiarity with these fields will minimize novelty based biases that may interfere with feedback.

## 4.2 Study Tasks

Each participant was asked to create a haptic experience for each of the scenarios described below (see figure 4) that they felt best matched the provided support narrative for the situation. Each participant was given five objects with corresponding descriptions, and asked to create a haptic experience for each of them. All objects were placed in a single VR scene and the participant was free to interact with these objects freely and create haptic scenarios in any order of their choosing.

1. Electric fence: A five meter tall metal fence is meant to be "electrified" to dissuade people from climbing over it. The fence has a single haptic space the covers the entirety of the fence area.

2. Rain cloud: On a calm day, a group of rain clouds produce light showers if walking under it. The rain clouds have a single haptic space covering the entire shadow of the cloud.

3. Tub and faucet: A tub with a faucet is mostly full of water with a running tap above it. Both the static water in the basin and the stream of water from the tap have their own haptic spaces.

4. Camp Fire: A pile of logs is burring with a fire that reaches waist height. The fire has a single haptic space that includes the entire area the fire travels during its animation.

5. Fake wall: Three brick walls block your path, but there is a hidden passage available by walking through the middle wall. Each of the solid walls on each side and the pass through wall in the middle have a haptic space.

## 4.3 Method

Before starting, participants were provided definitions for the terms "haptics", "haptic sensation", and "haptic experience" to ensure a common understanding of how these terms were being used in the study. Next, the participants were provided an outline of the controls and a description of how to access the menus, how to engage with an object to author the interaction, and what each slider affected.

Participants were then assisted with putting on the haptic vest, haptic gloves, and the headset as the test system was started. Once in the VR space, participants were asked to think aloud and describe their thoughts on the broader concept of immersive haptic authoring as they interact with our prototype system system. Starting with the electric fence experience, a description of the situation was provided (see above) and the participant was walked through creating a haptic sensation once. At this point the participant was free to proceed through creating at least one haptic sensation for each scenario. Once

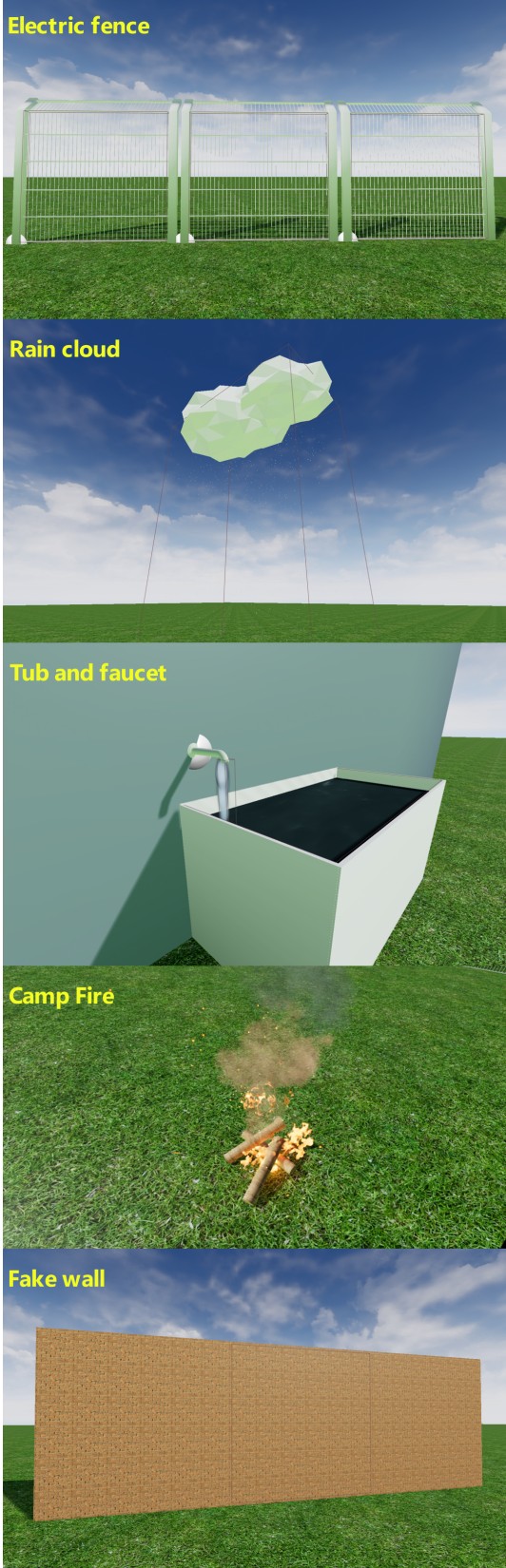

Figure 4: The five scenarios used in the study tasks in order from top to bottom: Electric fence, Rain cloud, Tub and faucet, Camp Fire, Fake wall

the participant indicated they were finished with all the scenarios, they removed all hardware and were asked a series of open ended questions about the considerations for the concept of immersive haptic authoring.

## 4.4 Post-study questionnaire

After creating all five situations, participants were asked a set of open ended question about their thoughts on an immersive haptic authoring approach to creating VR haptics, its potential, and its limitations. These questions included:

Q1. How would you describe what you just experienced?

Q2. Do you have any suggestions on what features a potential system could include that best leverage VR as an environment and why?

Q3. Are there any unique qualities to haptics design you believe a VR environment is better or worse suited for?

Q4.1. How do you foresee the scenario being designed for will impact this new authoring process?

Q4.2. How do you foresee the haptic devices being used will impact this new authoring process?

Q4.3. How do you foresee experience with the systems will impact this new authoring process?

Q4.4. Are there other factors, like the ones mentioned before, that you imagine will impact this new authoring process?

Q5. Do you have any thoughts on using an in-situ approach to haptic authoring related to creating haptic interactions versus haptic experiences?

Q6. Traditional desktop based haptic design requires editing a haptic interaction on a computer, loading the software to a headset, and experiencing the result and then returning to the desktop environment to iterate. How do you think an in-situ approach to haptic authoring will affect this and do you foresee any benefits or drawbacks?

If the participant was unclear on what the question was asking, it was repeated with different words but maintaining as closely as possible the same core question.

Once all interviews had been performed, the transcripts from the recorded audio was analyzed using open and axial coding [21][And cite textbook chapter].

## 5 RESULTS

The use of an immersive haptic authoring approach offers benefits to conceptualizing ideas though better feedback, removing motivation barriers to allow fast, convenient, and easy prototyping, and promoting a better understanding of the VR objects and world the haptics are being applied to. When considering what a future implementation may look like, a focus on designer freedom will have a strong effect on how designers approach the use of the system. If the system allows for physical manipulation, issues will also need to be addressed regarding effective navigation of the world. Each of these topics are discussed in greater details below with key quotes taken from the interviews to illustrate the concept.

## 5.1 Findings

### 5.1.1 Motivation

Motivation relates to the effect immersive haptic authoring will have on the will of a designer to create their haptic designs. The main results found matched our expectations and focus on the effects of reducing the need to move in and out of the VR space, and VR interaction technique's effect on unskilled haptic designers.

Immersive haptic authoring addresses many of the issues related to transitioning back and forth between a VR medium and a desktop medium during a haptic design process, which is a time consuming and disruptive act. By minimizing or removing the need to swap between a monitor and the VR headset, and remove haptic devices which are impractical to wear while using a computer, the amount of time overhead needed to iterate on a design is reduced. Reducing this overhead makes it more viable to iterate and refine small details and nuance rather than focusing only on high yield changes.

"It's so much better than having to go back to your computer screen, do the changes, come back, and be like, "no, that was wrong" and it just wastes so much time."

Reducing this jumping back and forth between VR and reality also reduces the context switching of thinking about the design from an experiential standpoint in VR and a design standpoint when outside VR. This can improve consistency and makes it easier to focus on an idea to implement.

"And I think that that's where it would be useful because then you don't have to break your context."

While all haptic designers can benefit from the reduced iteration time, unskilled haptic designers also benefit from a lower skill floor needed to implement a design. Because VR can utilize familiar interaction techniques such as using hands for selection, as well as presenting information in an intuitive and experiential manner, the amount of knowledge needed on how to use the authoring system can be reduced. It also makes it easier for an unskilled designer to form their idea through the use of trial and error to discover what they want. This is because haptics is ever present in real life but little attention is paid to it, so it is easier to recognize the correct solution than describe what the solution is.

"Being able to intuitively tell what something is rather than relying on numbers and sliders and dials and whatever, that lowers the barrier of entry because someone can go "yes, that's right"".

### 5.1.2 Authoring conceptualization

Authoring conceptualization relates to the effect immersive haptic authoring will have on the process of developing the haptic design to be implemented. This theme had the most findings of any phenomenon and focuses on how in context feedback affects idea exploration and how the scenario being designed constrains it.

One of the largest benefits of an immersive haptic authoring system is the ability to create a haptic design in the same context of VR it will be experienced by a user and shapes how to approach designing a scene's haptics. Defining the specific values a haptic interaction should have, may not reflect the sensation experienced when in the immersed context. By designing while in context, the evolution of the design is focused on what is actually being experienced, rather than what the designer hopes will be experienced.

"Say in isolation, the experience feels like it's 23 degrees Celsius. But in context, all of the users that you talk to say that it was really, really hot and weird and not what I (the designer) expected."

Feedback is also critical for ensuring that the end result ends up being experienced as it is conceptualized. The opportunities posed by an immersive haptic authoring approach such as faster and lower effort iteration increases how often feedback can be received. The high quality feedback from within VR both confirms if a desired effect has been achieved, but also provides prompting for new ideas to emerge. Even if the result is not as intended, by experiencing the result in the context being designed for it can both change the

designer's understanding of the haptics they are trying to create, and change the understanding of the context of the scene itself.

"I need to rapidly prototype, I need to iterate because I don't know what effect anything that I use is going to have necessarily. I have a basic intuition but that's all."

While an immersive haptic authoring approach does make it easier to explore ideas and prototype, not all scenarios make good use of this improved prototyping. Scenario complexity plays an important role in how deep a haptic interaction can be and influences how close base intuition gets to a desired effect. Scenarios that result in a dynamic feedback such as moving a paddle in a tub of water or waving a hand through a fire have the most to benefit. Because the haptic feedback is based on the specific action taken that produced it, feedback becomes more meaningful when these two events are observed simultaneously. In contrast if the scenario is simplistic such as the texture of a brick wall or feeling of an electric shock, intuition can start the design close enough to the desired result that only minimal iteration is needed. While this case is unlikely to fully eliminate the benefits of an immersive haptic authoring approach, it does greatly limit how effective the benefits can be and use of such an approach should be weighed against the limitations it imposes.

"As I mentioned, I think that interactive simulations are a very, very key one. If you have this sort of static signal, that's not going to dynamically change, then you can probably just prototype it at your desk."

### 5.1.3 Authoring tool expectations

Authoring tool expectations relates to abstract and specific utility that future implementations are expected to support by designers. The focus of discussion will placed on the abstract topic of designer freedom and not on specific design features which had many suggestions but there was insufficient overlap based on sample size to draw conclusions.

Designer freedom is important on multiple levels and allows the designer to feel empowered that they are able to use the authoring system as needed to meet their goals. One level of this freedom comes from being able to access a set of utilities that enable the design. While simple high level interfaces lead to many of the benefits discussed in other parts of this results section, it is likely impossible such a system will allow any design to be creatable. Even if the limitation is not reached, a system that has clear limitations shapes the haptic designs that will be considered which opposes the creativity and exploration an immersive haptic authoring approach encourages.

"That people are using this system really have to trust. I have to trust you, you are the designer of this system, in your letting me author haptic experiences but you're only giving me certain options and I have to trust that that's enough."

"Basically, you'd only have so much control as whoever the person designed it intended for you to have control."

Another level of freedom comes from being able to customize interfaces. As the set of tools needed can significantly vary based on scene, hardware, and intended effect, having irrelevant options can equally act as a distraction or as inspiration. Allowing the designer to make the choice on what they want to have access to at a given time helps them prioritize their focus on their design.

"For now I don't want this options with me, I want to change the layout of the VR so that it helps me to come up with the designs."

### 5.1.4 Physical manipulation

Physical manipulation relates to VR interactions in which the designer uses their hands as an interaction technique. Physical manipulation was mostly spoken in a negative context because of concerns for input precision, fatigue, and effective navigation of the world.

Because hand based systems are so similar to the way we interact with objects in the real world, expectations of similar levels of accuracy and detail are also created. Even when high precision is not needed, the expectation that touching an object or pointing at a spot will be exact remains, and failure to allow for high precision results in a sense of a lack of control in the designing process.

"It's not very precise because, I feel like I should be pressing a button and clicking"

The issue of precision also extends to characteristics beyond fidelity of the interaction method, and includes things like getting the correct viewing point to see where you are interacting with is also a factor in precise physical manipulation.

"For example, you're able to use that sense of depth in VR and place your hand further into a fire."

Physical manipulation also requires a significant amount of moving around. While it is reasonable to expect a designer to perform these motions for short sessions, designing in this way for entire work days is likely to cause fatigue. Even simple actions like needing to walk around an object or an interface design such as miniature model representations, instead of being able to move or rotate the objects itself, quickly become tiering when it must be done repeatedly over the course of a single edit.

"If I am creating a scene or trying out haptics then I'm constantly moving around, I'm standing up, I'm doing all this stuff, and that can be tiring to do for eight hours a day, every day."

Finally, using physical manipulation means a designer must be able to touch or accurately point to the object they wish to interact with. If the physical manipulation requires interacting with objects placed in the world, as opposed to menus which can be made to always be relative to the designer, the designer must move to these objects within the VR space. Navigating a VR world being designed in can be challenging, even with options to teleport to a location pointed at, as the location you want to go to may be a cityscape away or a completely different and unloaded level. This may be addressable by creating compartmentalized scenarios in which each scene a designer wishes to edit can be accessed in isolation, but still in full context, rather than engaging with it in the same world it will ultimately exist in.

"There's no way I'm walking an entire level, it's just, no. There's no chance of that happening. I would go to Unity."

### 5.1.5 VR presence

VR presence relates to the effects that being deeply involved within a scenario being designed for affects a designer. The main influences found were on understanding edited objects, and understanding of the editing scenarios.

While a designer has presence within the scene, they gain a better understanding of the objects they are trying to edit. This allows the designer to better understand the object's role and structure, and approach it as a real object rather than an abstract concept. This acts as a grounding point for the designer and helps reduce mental load of needing to recall past experiences with the object since it is now implicit while interacting with it.

"It brings you close to the objects themselves so you have a better sense of the actual object that you're going to be working with. There's a little bit less time spent imagining what the experience with the object will be because I can actually go up to the object that I'm going to work with."

VR Presence also affects a designer's understanding of the broader scene they are authoring haptics within. Through higher levels of presence within the scene, it becomes easier to understand the context of the haptics being designed for. This means the design can be done with consideration of other haptics, sounds, and objects already present in the scene, which is expected to produce a more desirable result.

"I do feel like I'm in the whole... I'm part of the scene." "If you're in a third person perspective on a computer designing things, you don't have that actual sensation of "okay, when I touch this object, it

should be this short and it should be. . . ", you're just removed from the design process."

## 5.2 Discussion

Based on the open coding analysis, we believe immersive haptic authoring can be used to improve the process for designers to add haptics to virtual environments. While we initially considered a workflow solely in VR would be ideal, participant discussion consistently suggested a mixture of VR and traditional desktop interfaces would be necessary out of concern that the immersive haptic authoring system would lack the flexibility or accuracy needed for some tasks. While it may be possible to address these concerns and allow a full VR workflow, it may not be necessary. There was no sense in the discussions that needing to periodically leave the VR space was considered a negative if the majority of the design process could be done in VR. We believe as long as the need to leave an immersive haptic authoring system is minimally intrusive, such as occurring between design iterations, and infrequent, suggesting upwards of 15 minute periods within VR, immersive haptic authoring will play a complementing role to existing haptic authoring processes.

At a theoretical level, this authoring approach has potential for improving time efficiency, simplifying workflow, and improving accessibility for authoring haptics. The haptics created using immersive haptic authoring are expected to be of higher quality due to the improvement in a designer's understanding of the wider contextual environment being designed in, providing a clearer representation of VR object directly being designed for, and providing easier and faster feedback on prototyped iterations. These topics were found to be the most important and impactful, and should be of specific note for future work when creating a comprehensive theory on an immersive haptic authoring system's place in design.

When considering specific implementations that may follow, ensure a high level of user freedom beyond what is strictly needed for a situation is by far the most important aspect to potential designers. While occasional comments alluded to the benefits of specialized systems for a specific use or piece of hardware, the concern that the immersive editor would imply a restriction in tools and options was far stronger. If the editor system is looking to be applicable to practical larger scale projects, a form of efficient world scale movement will also need to be used to reach distinct areas of the world. Finally, while allowing physical manipulation as an interaction medium brings designers closer to the objects they are editing, its difficulties with precision and likely-hood of fatigue suggest it will need to be coupled with other interactions to support it.

## 6 LIMITATIONS AND FUTURE WORK

Our look into immersive haptic authoring show the beginnings and possibilities this approach offers, and demonstrates the potential for further research. The three main areas of future work are as follows.

The user study performed was done with a small sample size and although every participant had backgrounds in VR or haptics, only one had a strong background in both. A larger study that focuses on professionals specialized in VR haptic design work would be able to extend the findings presented in this paper and provide a more confident assertion on the practical concepts of immersive haptic authoring. This may be difficult though as haptic design for VR is not a common job making finding suitable participants difficult.

While our work contributes many key ideas and considerations for immersive haptic authoring, it stopped short of creating a unified theory. The grounded theory analysis done on the transcript data only included open and axial coding to draw initial connections between ideas. With a larger and stronger data set, selective coding should also be done to create a explicit theory of use for immersive haptic authoring.

Finally, A more robust implementation that expands the functionality and allows more complex interaction concepts such as the ability to create dynamic effects that are localized to a point of contact and allowing haptic interactions to be defined as changing over period of time or through space will allow a more direct analysis of an immersive haptic authoring approach in a practical setting. This would explore the space more thoroughly and help to understand if the concepts scale to the demands more complex haptic interactions require in realistic situations.

## 7 CONCLUSION

Virtual reality is a quickly growing technology and as people begin to expect more immersion and interaction from it, more interest and demand in haptics is likely to follow. The concepts we present for creating immersive haptic authoring systems show the potential of this approach to improving the haptic interaction prototyping process with the goal of creating deeper and more impactful interaction. An evolution in technology needs an evolution in how to interact with the technology, and we have shown moving the authoring process of haptics into the same environment as it will be experience in has potential advantages. The user study found motivation, authoring conceptualization, authoring tool expectations, physical manipulation, and VR presence as areas of importance, and suggests the need for further research in these areas. For the very practical need being created for haptics in VR, this work presents a novel approach to the problem and a foundation for further research work to build on.

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
