# OpenReview forum: "Immersive Haptic Authoring: Creating Haptic Experiences On-Demand within VR through Direct Manipulation"
_graphicsinterface.org/Graphics_Interface/2023/Conference — Submitted to GI 2023_

### Official Review · Reviewer_KWmK · 2023-01-13
**Slightly positive**

**Rating:** 7
**Confidence:** 2

**Review:**

In this manuscript, the authors present their design goals, implementation, and evaluation of a prototype to propose authoring haptic experiences directly in VR instead of a computer. The paper is quite relevant to GI although a conference like IEEE VR or ISMAR seem to be even more relevant for this kind of work.

It is an easy and timely read, there are some typos in the manuscript (see below for a non-exhaustive list) and some sentences are sometimes quite difficult to parse. I would argue that the contribution of the authors are clear and that the length of the manuscript correlates well to that of the strength and number of the contributions.

I am not really an expert in authoring of haptic, or in haptic experiences as a whole, I therefore rate myself as a 2 out of 5 on the expertise/confidence in my review. That being said, I think that the authors provided an easy to read background section, although the section (and maybe the paper) would benefit from more descriptions and links to professional authoring solutions. By this, I mean that naming a few of them and describing the habitual workflow of designers with such tool to provide an easier comparison with the tool that they provide would surely strengthen the manuscript.

I would have loved to see the code of the prototype being released on a github repository for others to use in their research too. It seems that what the authors are developing can really be useful to experts and I find it sad that it cannot be reused by them. While I would understand that the authors think that their code might not be publication ready yet, I would still argue that it does not really matter: they can put a link to a github repository in the manuscript and continue cleaning the code after submission/publication of the paper.

Overall, I am pretty positive about the paper and I think it presents an interesting contribution to the community. However, as I am not an expert, there might be things that I have missed and I look forward to discussing this further with other reviewers.

typos:
- enviornment (page 1)
- page 2 --> unfortunate phrasing "researched. Research"
- "will be take"

---

### Official Review · Reviewer_wGAa · 2023-01-17
**Useful direction but work is in early stages**

**Rating:** 3
**Confidence:** 4

**Review:**

This paper presents an immersive authoring tool for creating haptic feedback in VR experiences.

This research takes a useful and relevant direction toward designing haptic feedback for VR. Overall, the work as presented is fairly preliminary and is more like a poster contribution than a full research paper.

I would not consider the use of immersive authoring for designing haptics as a 'novel concept', nonetheless, it is a topic worth exploring. The current approach is quite limited however. I would like to see this work taken further to identify the real potential for an immersive interface, that takes advantage of the benefits of VR.

A strong limitation of the current tool is that it only allows users to select pre-defined objects for haptic interaction. This can be done quite easily with a mouse click, whereas an immersive interface may allow more complex 3D regions to be defined more easily. The haptic interaction is also very straightforward with only 2 parameters for strength and duration. I believe that designers could get the gist of their implementation using a controller with a desktop monitor in this case. Again, it would be interesting to explore how to support the creation more complex haptic interactions, for instance that vary with time or in strength over a region. Perhaps the haptic effects could have more textured spatial or temporal patterns. Such spatial and temporal complexity could help motivate the need for an immersive tool.

The approach for designing such a tool would benefit from a strong user elicitation study to define the system requirements. Who are the users and what kinds of things would they like to accomplish? What do they currently have difficulty doing with a desktop interface?

As acknowledged in the paper, 6 participants is a very small sample size. I agree it may be enough to provide some insightful feedback, but there is limited insight to be gained from such a straightforward implementation. I find the results quite lengthy and in need of some stronger takeaways. I also advise to be careful making strong claims that are not supported by the evidence, for instance that "the amount
of time overhead needed to iterate on a design is reduced". Such a finding would require a direct comparison with a 2D interface. I don't suggest that such a comparison would necessarily be the most likely way to produce interesting insights, but claims about such differences should be avoided.

I encourage the authors to continue their work. I believe there is much to explore here and value to be found in a more advanced tool.

---

### Official Review · Reviewer_27pz · 2023-01-19
**Unclear contribution due to missing prior work**

**Rating:** 3
**Confidence:** 5

**Review:**

The authors present an in-VR design tool to help design tactile effects for a haptic body suit, and evaluate it with 6 experts.

I recommend reject. While haptic design tools are an important topic, and in-VR design an interesting new direction within the literature, this paper does not demonstrate awareness of the rapidly increasing haptic design and design tool literature, including two systems that already propose in-VR editing. As such, the novelty of this work is not convincing, and many claims (e.g., the limitations of desktop-based editors) are not supported. In addition, I believe the paper lacks polish and could benefit from improvements in presentation.

Limited related work: this paper references only a subset of the haptic design and design tool literature, including critical prior work. Most important are VRPlay [a] and VRTactileDraw [b]. Both of these systems employ an extremely similar interaction method. The parameterized editing brings to mind Feel Effects [c], a much more rigorous investigation of parameterized haptics that may inform a richer discussion in this system. Similar desktop design tools that may need to be considered are Tactile Animation [d] and TactJam [e,f] is a relevant citation as both contain a similar multi-actuator spatial editor, albeit on a desktop [d] or desktop and embedded interface [e,f]. Finally, I find the criticism of desktop-based editing has potential, but isn't quite convincing; using references to [a,b] or work into haptic design processes [g,h] may help motivate the paper's points. There are additional papers that might be worth considering, including an early demonstration-based editor for haptics [i].

Presentation issues: I found that many of the points made were a little unconvincing, and written in a colloquial style that doesn't seem to fit GI as a venue. There are missing references, including a TODO about a textbook; the whole paper needs a bit of polish and crystallization of its points.

Positives: The study methodology seems fine to me, and the tool does seem to have some novelty (e.g., combining triggers with parameterized haptic effects). However, without situating the work more firmly in the related work, the novelty and contribution are simply not clear.

Citations
[a] Huang DY, Chan L, Jian XF, Chang CY, Chen MH, Yang DN, Hung YP, Chen BY. Vibroplay: Authoring three-dimensional spatial-temporal tactile effects with direct manipulation. InSIGGRAPH ASIA 2016 Emerging Technologies 2016 Nov 28 (pp. 1-2).
[b] Kaul OB, Domin A, Rohs M, Simon B, Schrapel M. VRTactileDraw: A Virtual Reality Tactile Pattern Designer for Complex Spatial Arrangements of Actuators. InIFIP Conference on Human-Computer Interaction 2021 Aug 30 (pp. 212-233). Springer, Cham.
[c] Israr A, Zhao S, Schwalje K, Klatzky R, Lehman J. Feel effects: enriching storytelling with haptic feedback. ACM Transactions on Applied Perception (TAP). 2014 Sep 29;11(3):1-7.
[d] Schneider OS, Israr A, MacLean KE. Tactile animation by direct manipulation of grid displays. InProceedings of the 28th Annual ACM Symposium on User Interface Software & Technology 2015 Nov 5 (pp. 21-30).
[e] Wittchen D, Fruchard B, Strohmeier P, Freitag G. TactJam: a collaborative playground for composing spatial tactons. InProceedings of the Fifteenth International Conference on Tangible, Embedded, and Embodied Interaction 2021 Feb 14 (pp. 1-4).
[f] Wittchen D, Spiel K, Fruchard B, Degraen D, Schneider O, Freitag G, Strohmeier P. TactJam: An End-to-End Prototyping Suite for Collaborative Design of On-Body Vibrotactile Feedback. InSixteenth International Conference on Tangible, Embedded, and Embodied Interaction 2022 Feb 13 (pp. 1-13).
[g] Schneider O, MacLean K, Swindells C, Booth K. Haptic experience design: What hapticians do and where they need help. International Journal of Human-Computer Studies. 2017 Nov 1;107:5-21.
[h] Seifi H, Chun M, Gallacher C, Schneider O, MacLean KE. How do novice hapticians design? A case study in creating haptic learning environments. IEEE Transactions on Haptics. 2020 Jan 23;13(4):791-805.
[i] Hong K, Lee J, Choi S. Demonstration-based vibrotactile pattern authoring. In Proceedings of the 7th International Conference on Tangible, Embedded and Embodied Interaction 2013 Feb 10 (pp. 219-222).

---

### Meta-Review · Area_Chair_Wa4M · 2023-01-22

**Recommendation:** 4
**Confidence:** 4

**Metareview:**

In this manuscript, the authors present an in-VR design tool to help design haptic experiences.

All reviewers agree that the work is interesting and relevant to the conference and would be a good fit. However, it seems to early to be presented at the conference in the current state. Indeed, it seems that the authors have not included enough related work to put their work in context and to show the novelty of the work and the contributions. The reviewers also seem to agree that there is not enough thoughts and research onto the design of the tool.

Overall, while it is an interesting paper, it currently lacks a clear-cut contribution with respect to prior work and more justifications for the design of the tool as well as a stronger evaluation.

I wish luck in the authors' work on this idea which is promising.